# THE WEIGHTED MEAN TRICK –
# OPTIMIZATION STRATEGIES FOR ROBUSTNESS

## ABSTRACT

We prove that minimizing a weighted mean results in optimizing the higher-order moments of the loss distribution such as the variance, skewness, and kurtosis. By optimizing the higher-order moments, one can tighten the upper bound of the loss mean deviating from the true expectation and improve the robustness against outliers. Such types of optimization problems often lead to non-convex objectives, therefore, we explore the extent to which the proposed weighted mean trick preserves convexity, albeit at times at a decrease in efficiency. Experimental results show that the weighted mean trick exhibits similar performance with other specialized robust loss functions when training on noisy datasets while providing a stronger theoretical background. The proposed weighted mean trick is a simple yet powerful optimization framework that is easy to integrate into existing works.

## 1 INTRODUCTION

The most common objective in machine learning is to minimize the average loss. However, by doing this it implies that all samples are equal. The goal of this work is to convince the reader that from the perspective of the model, not all samples are equal and optimizing a weighted mean is a more progressive approach. For instance, at the end of the training, few samples are left uncaptured by the model which yield large loss values. One might choose to put more weight on those samples to help the model learn (individual fairness), or consider them noise and assign them less weight (robustness). The decision will also be reflected by the loss variance; it will decrease when hard samples are weighted more and increase otherwise.

Before introducing more formally the weighted mean, we first justify theoretically the impact of variance penalization. Using the empirical Bernstein bound (Maurer & Pontil, 2009) for i.i.d. loss values $Z$ and bounded variance we have with probability $1 - \delta$:

$$\mathbb{E}[Z] - \frac{1}{n}\sum_{i=1}^{n} Z_i \;\leq\; C_1 \sqrt{\frac{2\mathbb{V}_n[Z]\ln 2/\delta}{n}} + C_2 \frac{7\ln 2/\delta}{3(n-1)} \tag{1}$$

where $C_1$, $C_2$ are problem depended constants. This inequality reveals two things. First, we have with high probability that the empirical mean is close to the theoretical value. This bound along with similar PAC-Bayes bounds (Seldin et al., 2012; Tolstikhin & Seldin, 2013) justifies the practical success of the empirical risk minimization (ERM) which has the objective to minimize the mean loss value (Namkoong & Duchi, 2017). Secondly, the difference between the two is bounded in terms of the empirical variance. This second implication led to a large and growing body of studies that investigate variance penalization (Maurer & Pontil, 2009; Namkoong & Duchi, 2017; Duchi & Namkoong, 2019; Staib et al., 2019; Lam, 2019; Heinze-Deml & Meinshausen, 2021; Hu et al., 2018). Moreover, by penalizing the variance, the intrinsic bias-variance tradeoff of the ERM can be controlled. Two studies influential for this paper are that of Duchi & Namkoong (2019) and Li et al. (2021). Duchi & Namkoong (2019) proposed taking the expectation with respect to a different distribution which allowed penalizing variance while preserving the convexity of the objective. Similarly, Li et al. (2021) investigated optimizing a tilted empirical risk which is equivalent to penalizing all the higher-order moments simultaneously. In summary, previous methods either penalize only one moment or all the higher-order moments but are not flexible enough to penalize any desired combination of the higher-order moments.

Inspired by the above works, we propose to optimize a weighted mean and prove that for certain weights, it is equivalent to optimizing any higher-order moments of the loss distribution such as variance, skewness, and kurtosis. Our approach generalizes that of Duchi & Namkoong (2019); Li et al. (2021) while simplifying the optimization procedure and enabling separate penalization of the higher-order moments.

In particular, we will construct the weights $w$ such that optimizing the weighted mean of the loss $\ell$ is equivalent to applying a variance penalization:

$$\mathbb{E}[w\ell] = \mathbb{E}[\ell] + \lambda \mathbb{V}[\ell] \tag{2}$$

or penalization of other central moments (e.g., skewness, kurtosis). Construction of these weights are treated by Theorems 1 and 3 from section 3, which require minimal computational resources. Important note, penalizing directly the variance preserves convexity only for values of $\lambda$ within a very narrow range (Maurer & Pontil, 2009). However, the weighted mean method yields a convex objective for any positive values of $\lambda$.

In detail, our contributions are as follows:

(**C1**) We build upon the work of Duchi & Namkoong (2019) and prove that optimizing a weighted mean is equivalent to reducing or amplifying both variance (Theorem 1) and higher-order moments (Theorem 3).

(**C2**) We derive the limits of the penalization interval which preserves convexity (Lemma 2) and also show how to penalize with values outside this interval while still maintaining convexity (Lemma 4).

(**C3**) We connect the variance and higher-order moments penalization using weighted mean to the robustification of loss functions (Lemma 5 and Lemma 6).

(**C4**) We develop a convex version of the variance penalized cross-entropy loss which provides a higher accuracy in high noise scenarios with class dependent noise.

(**C5**) We show experimentally that a negative variance penalization improves model accuracy when training with noisy labels.

The implications of the weighted mean are much broader than what we investigate in this work. We limit the scope of this paper to classification using deep neural networks trained with noisy labels. But the mathematical framework also covers the control of the bias-variance trade-off and can also be applicable to regression problems.

In the sequel, we start by introducing the notation in section 2, then in section 3 we present the moment penalization strategy and the weighted mean trick which is a computational technique to make moment penalization practical. Next, in section 4, we illustrate the application of moment penalization using the weighted mean formulation to optimize for robustness when training with noisy labels.

## 2 NOTATIONS

In subsequent sections we will use the following notation. A training data set is defined as $\mathcal{D} = \{(x_i, y_i)\}_{i=1}^n$ where $x_i \in \mathcal{X}$ are the features and $y_i \in \mathcal{Y} = \{1, \ldots, k\}$ represent the class labels. A classifier $f(x; \theta)$ is a mapping $f : \mathcal{X} \times \Theta \to \mathcal{V}$ from the feature space to the probability simplex $\mathcal{V}$ parameterized by $\theta$. A loss function $\ell : \mathcal{V} \times \mathcal{Y} \to [0, \infty)$ gives a penalty $\ell(v, y)$ when the model predicted the value $v$ and label $y$ was observed. A weight function $w : \mathcal{V} \times \mathcal{Y} \to \mathbb{R}$ assigns to each sample a weight. As we are interested in optimizing the model output value $v$ that minimizes the penalty, we will focus our investigation on loss functions $\ell(v, y)$ that are convex in $v$ and seek to preserve the convexity when optimizing the weighted mean. In ERM, we are interested with finding the model parameters $\theta$ that minimize the empirical risk calculated as $\mathbb{E}_{\mathcal{D}}[\ell(f(x_i; \theta), y_i)]$ and the weighted form $\mathbb{E}_{\mathcal{D}}[w(\ell(f(x_i; \theta), y_i))\ell(f(x_i; \theta), y_i)]$. To simplify the notation, we will drop the dataset $\mathcal{D}$ from the expectation subscript and the arguments of the loss and the weight function, i.e., $\mathbb{E}[\ell] \Leftrightarrow \mathbb{E}_{\mathcal{D}}[\ell(f(x_i; \theta), y_i)]$ and $\mathbb{E}[w\ell] \Leftrightarrow \mathbb{E}_{\mathcal{D}}[w(\ell(f(x_i; \theta), y_i))\ell(f(x_i; \theta), y_i)]$. Similarly, the notation for minimum is simplified as $\min \ell \Leftrightarrow \min_{x_i, y_i \in \mathcal{D}} \ell(f(x_i; \theta), y_i)$.

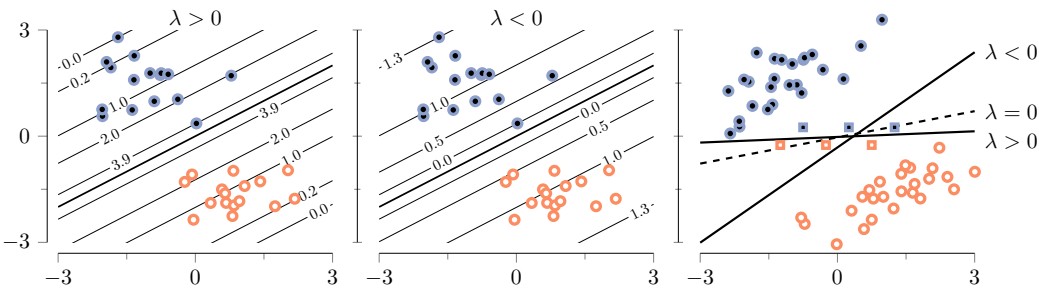

Figure 1: Variance penalization for classification problems. Contour lines of the weights distribution for positive $\lambda$ are shown on the left and for negative on the center plot. The right plot shows the use of variance penalization for outlier suppression or amplification (robustification versus generalization).

## 3 PENALIZING MOMENTS

In ERM, the impact on the model parameters of a sample is determined by the balance between its loss value and the average of the training batch. Especially observed in the latter stages of the training when majority of samples have a small loss value except few which are left uncaptured by the model. Case in which we consider them as unlearned samples and amplify their impact or as outliers and suppress their impact on the model. The *weighted mean trick* consist in assigning weights to each sample based on the loss value to either amplify or suppress their impact on training. This is similar to dropout (Hinton et al., 2012) but for losses. However, *weighted mean trick* is a deterministic process and the weights are not restricted to 0 or 1 but can take any non-negative value.

In what follows, we apply the *weighted mean trick* to extend the ERM framework to multi-objective optimization of the mean and higher-order moments of the loss function. First, we show that for some distinct weights, optimizing the weighted mean is equivalent to a simultaneous optimization of the mean and variance.

**Theorem 1** (Variance Expansion). *Let $\ell$ be a loss function with finite $\mathbb{E}[\ell]$ and finite $\mathbb{V}[\ell]$ and let $w(v, y) = 1 + \lambda(\ell(v, y) - \mathbb{E}[\ell(v, y)])$, then we have:*

$$\mathbb{E}[w\ell] = \mathbb{E}[\ell] + \lambda\mathbb{V}[\ell] \tag{3}$$

*Proof.* Replacing $w$ with the definition and using the linearity property of the expectation along with Proposition 7 we get: $\mathbb{E}[w\ell] = \mathbb{E}[\ell] + \lambda\mathbb{E}\big[(\ell - \mathbb{E}[\ell])\ell\big] = \mathbb{E}[\ell] + \lambda\mathbb{E}\big[(\ell - \mathbb{E}[\ell])^2\big] = \mathbb{E}[\ell] + \lambda\mathbb{V}[\ell]$ ☐

Thus, switching from mean to weighted mean allows us to control the bias-variance tradeoff through $\lambda$ to improve the distributional robustness of the model (Maurer & Pontil, 2009). However, the range of $\lambda$ values that preserve the convexity of the objective depends on the average and the minimum penalty returned by the loss function $\ell$ as shown by the next lemma.

**Lemma 2.** *As introduced in Theorem 1, the variance expansion of a convex loss function $\ell(v, y)$ in $v$ yields a new objective that is also convex in $v$ if $\lambda \in [0, \lambda_{max}]$, where $\lambda_{max} = 1/(\mathbb{E}[\ell] - \min \ell)$.*

Proof is provided in Appendix A. This lemma precisely shows why directly penalizing the variance does not preserve convexity besides when $\lambda$ takes values in a narrow interval. Moreover, directly penalizing the variance as part of a numeric optimization objective, the upper limit of the interval, $\lambda_{max}$, is not constant and changes with each iteration of the optimization algorithm. Note that the further the minimum value $\min \ell$ is from the average loss value $\mathbb{E}[\ell]$, the narrower the interval is. Conversely, when $\mathbb{E}[\ell] = \min \ell$ results that $\ell$ is constant and $\mathbb{V}[\ell] = 0$ thus the objective is convex for $\lambda > 0$. However, using the weighted mean trick the objective remains convex for any positive $\lambda$ irrespective if $\mathbb{V}[\ell] = 0$.

Figure 1 shows the weights distribution for different values of $\lambda$. When $\lambda$ is positive (left plot) samples closer to the decision boundary (which also means with larger loss values) receive more weight. On the other hand, when $\lambda$ is negative (center plot) samples with larger loss values receive

less weight. Of note, for $\lambda < 0$ the objective is not convex, however as we will show later will still converge to an optimal solution. The right plot shows a binary classification problem where each class consists of two clusters, squares and circles. The parameter $\lambda$ controls the placement of the decision boundary with respect to these two clusters. Positive $\lambda$ values place more weight on the cluster of squares which is closer to the decision boundary and as a result the boundary is horizontally aligned. On the other hand, negative $\lambda$ values place more weight on samples farther from the boundary and in this case the cluster of circles. Therefore, this aligns the decision boundary with respect to the cluster of circles oriented diagonally.

Variance is not the only central moment that can be penalized using the weighted mean. In fact, any combination of the moments can be penalized. The next result generalizes Theorem 1.

**Theorem 3** (Moments Expansion). *Let $\ell$ be a loss function with finite first $m$ central moments and define $w(v,y) = \sum_{i=1}^{m} \lambda_i (\ell(v,y) - \mathbb{E}[\ell(v,y)])^{i-1}$, then we have:*

$$\mathbb{E}[w\ell] = \lambda_1 \mathbb{E}[\ell] + \sum_{i=2}^{m} \tilde{\lambda}_i \mathbb{E}\left[(\ell - \mathbb{E}[\ell])^i\right] \tag{4}$$

*where $\tilde{\lambda}_i = \lambda_i + \lambda_{i+1} \mathbb{E}[\ell]$ for $i < m$ and $\tilde{\lambda}_m = \lambda_m$*

Proof is provided in Appendix A. We notice that penalizing moments higher than two incurs an additional penalization of the previous moment. For example, penalizing skewness by $\lambda_3$ incurs a variance penalization of $\lambda_3 \mathbb{E}[\ell]$.

Theorem 3 also has an algebraic interpretation. Note that the formula for weights is nothing more than a polynomial in $\ell(v,y)$ translated by $\mathbb{E}[\ell]$. When penalizing the variance, $\lambda_2$ controls the slope of the linear equation, and when penalizing the skewness, $\lambda_3$ controls the curvature of the quadratic equation. Moreover, the penalization factors $\lambda_i$ also define the placement of the roots of the polynomial and the convexity of the weighted mean objective.

**Lemma 4** (Convexity of Moments Expansion). *Let $\ell(v,y)$ be a loss function convex in $v$ and $p : \mathbb{R} \to [0, \infty)$ be a non negative and differentiable convex function and $M \geq 0$, then the weighted objective $w(v,y)\ell(v,y)$ with $w(v,y) = p(\ell(v,y) - M)$ is convex in $v$ if $p$ is non decreasing.*

Proof is provided in Appendix A. The above result lists two requirements on the polynomial function $p$ such that the weighted mean objective remains convex. First, the function must be non-negative, thus, the negative weights must be clipped to 0. Secondly, when the function takes positive values, $p$ must be non decreasing and convex. Of note, this result is similar in scope to Lemma 2 but does not generalize it as the weights are non negative. Figure 2 shows several examples of polynomials. Variance penalization implies $p$ is an affine function, and thus, convex. However, for $\lambda_1 > 0$ the polynomial is non decreasing (left plot) and for $\lambda_1 < 0$ the polynomial is non increasing (center plot). Thus, only positive variance penalization will result in a convex objective. Of note, Lemma 2 upper bounds $\lambda_1$ if weights are not clipped to 0, however, if clipping is used $\lambda_1$ is not upper bounded. When penalizing skewness, $p$ is a quadratic function (right plot). In this case, if $\lambda_2 > 0$ then $p$ is convex and non decreasing only on a restricted interval instead of the entire real line. Thus, convexity can be preserved by appropriately adjusting the roots of the polynomial such that $p$ is convex on $[\min \ell - \mathbb{E}[\ell], +\infty]$.

Clipping negative weights to 0 prevents the optimization objective to switch from minimization to maximization, usually an undesirable behavior. Consequently, the samples with a corresponding zero weight reach their maximum contribution in the moments penalization. Further increasing the factors $\lambda_i$, will have no effect on those samples and only samples with non-zero weights will participate in the training. As a result, the efficiency of the moments penalization will slightly fall. Moreover, the samples with zero weight will be excluded from training, technique used in the past by multiple studies. Rockafellar & Uryasev (2000) used a similar clipping technique to optimize only for samples that are part of the tail of the distribution.

The Moments Expansion Theorem extends the variance expansion proposed by Duchi & Namkoong (2019) to include higher-order moments. However, even when only the variance is penalized, the two methods are still slightly different. Moments Expansion Theorem computes the weights directly from the loss values, whereas the variance expansion of Duchi & Namkoong (2019) solves a secondary optimization problem to find the weights. The use of a secondary optimization problem has the

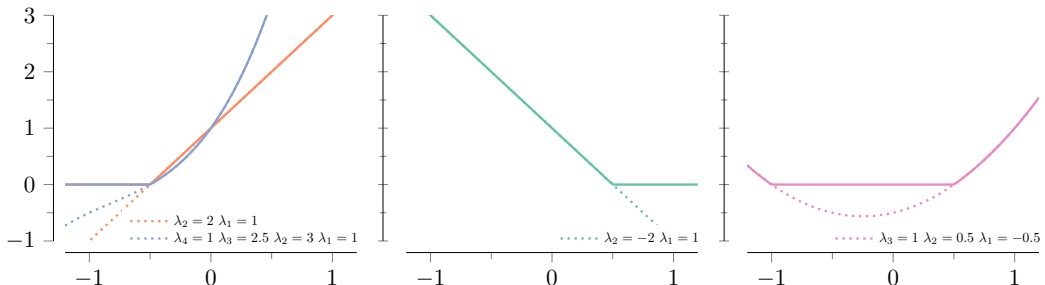

Figure 2: Polynomial functions for moments penalization. Dotted lines show the complete polynomial whereas solid lines the clipped version. Left plot shows two convex and non decreasing polynomials. Center plot shows a convex but non increasing polynomial and thus will result in a non convex objective. Right plot shows a convex and non decreasing polynomial on the interval $[-1, +\infty]$, and thus will yield a convex objective when $\min(\ell) - \mathbb{E}[\ell] \geq -1$.

advantage of penalizing the variance more consistently despite being more computationally expensive. On the contrary, when using the moments expansion, the penalized variance can be slightly lower depending on the number of weights that are 0. However, the advantage is that the direct computation of the weights makes it easier to include the method into existing analysis frameworks. A similar method that extends the optimization objective to include penalization factors for higher-order moments was proposed by Li et al. (2021). The proposed method replaces the ERM objective with a tilted version calculated as $\frac{1}{t}K(t) = \frac{1}{t}\log\mathbb{E}[e^{t\ell}]$ where $K(t)$ is the cumulant-generating function of the loss $\ell$. The penalization of the higher order moments of the tilted ERM can be recovered from the power series expansion of $K(t)$. The distinction between the two is that the moments penalization introduced in this paper represents a generalization of the tilted ERM as it allows any combination of the higher order moments to be penalized whereas tilted ERM uses a single parameter that governs the penalization factors. In summary, the moments penalization implemented using the weighted mean trick is more flexible, however, it comes at a cost as there are more parameters to tune when penalizing multiple moments compared to tilted ERM of Li et al. (2021).

**Convergence and convergence rates.** The moments penalization problem along with variance expansion of Duchi & Namkoong (2019) fall under the class of distributional robust stochastic programsSun & Xu (2016) (DRSP) which is a subclass of ambiguity programsRoyset & Wets (2017) (AP) where the general objective is:

$$\text{AP:} \quad \min_{\theta \in \Theta} \sup_{P \in D(\theta)} \varphi(\theta, P) \tag{5}$$

where $\Theta$ is the set of model parameters and $D(\theta)$ is the ambiguity set. In DSRP, the bivariate function $\varphi(\theta, P) = \mathbb{E}_P[\ell]$ where $\ell$ is the loss function and the ambiguity set $D(\theta)$ is a set of probability distributions. In the case of moments penalization problem, the ambiguity set $D(\theta)$ depends on the model parameters and is a singleton as the weights uniquely transform the empirical distribution. As a result, the optimal value of the inner maximization problem becomes $\sup_{P \in D(\theta)} \varphi(\theta, P) = \mathbb{E}_P[\ell]$. *Intuitively, a model will converge if changes in its parameters will cause minor changes in the distribution $P$, and with each step the distribution will approach the optimum distribution $P^*$.* Formally, to quantify the changes in the distribution, we would need a distance or a metric. Sun & Xu (2016) use the total variation metric and a pseudometric to prove uniform convergence, possibly at an exponential rate, if $P$ converges to $P^*$ under total variation metric and $\ell$ is uniformly bounded (see Sun & Xu, 2016, Th. 1 and Prop. 3). Royset & Wets (2017) proposed a hypo-distance metric and proved lop-convergence given that the bivariate function $\varphi(\theta, P)$ satisfies some assumptions, (see Royset & Wets (2017) Def. 4.1). Duchi & Namkoong (2019) provide guarantees for a number of stochastic risk minimization problems when only the variance is penalized and $P$ is in the local neighborhood of the empirical distribution defined using the $\chi^2$-divergence. We refer the reader to the works of Sun & Xu (2016) and Royset & Wets (2017) and the references therein for additional guarantees if more information about the problem structure is available, or if other metrics are used. For the moments penalization problem, the moments penalization factors $\lambda_i$ for $i \geq 2$ determine how

much the distribution $P$ changes when the loss changes. Small values of the penalization factors will keep $P$ in the neighborhood of the empirical distribution, whereas large values will make the weights sensitive to changes in the loss values that can cause stability or convergence issues. The exact values depend on the empirical distribution of the data and the choice of the model and loss function.

**Weighted mean trick in practice.** To apply the method in practice, the classical batch training algorithm must be extended to include an additional step, the weights calculation. Instead of directly calculating the average loss, the user will calculate the loss value for each sample in the batch and then use the expression from Theorem 3 to compute the weights and the weighted mean. The moments penalization factors $\lambda_i$ are the hyper-parameters and are tuned in ascending order with respect to $i$. However, penalizing higher-order moments might affect the impact of the lower-order ones, and thus, it might require a few iterations to find the optimal combination. The implementation of this algorithm in Pytorch (Paszke et al., 2019) is available on GitHub.[1] Since the gradient of the weighted mean is the weighted gradient of the elements, this allows weights to control the impact of each sample on the model parameters. Of note, the theoretical results hold when switching from expectation to sample expectation, $\mathbb{E}_n[\ell] = \frac{1}{n} \sum_{i=1}^{n} \ell(f(x_i, \theta), y_i)$.

---

**Algorithm 1:** Training with Moments Penalization

**input :** 
$f(x; \theta)$ – model to be trained
$\{x_i, y_i\}_1^n$ – batch of training data
$\{\lambda\}_1^m$ – penalization factors
$\ell(v, y)$ – loss function

**while** *stopping criteria not reached* **do**
  **for** $i \leftarrow 1$ **to** $n$ **do**
    |  $z_i \leftarrow \ell(f(x_i, \theta), y_i)$ ;               /* sample loss */
    $w \leftarrow \left[ \sum_{j=1}^{m} \lambda_j (z - \mathbb{E}_n[z])^{j-1} \right]_+$
    $\mathcal{L}_w \leftarrow \frac{1}{n} \sum_{i=1}^{n} w_i z_i$ ;           /* weighted mean */
    $\theta \leftarrow \theta - \gamma \nabla_\theta \mathcal{L}_w$ ;       /* update model parameters */

---

## 4 ROBUST CLASSIFICATION

In this section, we will explore the robustification of the cross-entropy function under label noise by bounding the loss values using negative variance penalization. In this case, the resulting objective will not be convex. However, for small penalization factors the objective remains convex on almost the entirety of the domain (Figure 3) and does not hinder the convergence. Moreover, we also develop a convex version of the variance penalized cross-entropy though for a minor price in performance.

The objective of robust classification is to learn an optimal classifier $f^*$ that minimize the average loss for both clean and noisy data. Formally, $f^* = \arg\min \mathbb{E}[\ell(f(x, \theta), \hat{y})]$ where $\hat{y}$ represents noisy labels. The noise considered in this paper corrupts the class labels with probability $\mathbb{P}(\hat{y}_i \neq y_i) = \eta$ or preserves it with probability $\mathbb{P}(\hat{y}_i = y_i) = 1 - \eta$, where $\eta \in [0, 1]$. We investigate two scenarios: when $\eta$ does not depend on the class label (class independent or uniform noise), and when $\eta$ depends on the class label (class dependent or asymmetric noise). In what follows, we show that using a negative variance penalization factor, $\lambda_2 < 0$, bounds the loss function. By bounding the loss function the impact of misclassification of noisy samples on the average loss will decrease, and thus reduce the noise impact on the model during training.

Ghosh et al. (2017) outlined the distribution independent sufficient conditions for a loss function to be robust under both, class dependent and class independent noise. Specifically, if the loss function satisfies the symmetry constraint:

$$\sum_{i=1}^{k} \ell(f(x, \theta), i) = C \tag{6}$$

where $k$ is the number of classes, $C$ is a constant, and the equality holds $\forall f, \forall \theta, \forall x \in \mathcal{X}$, then $\ell$ is noise tolerant for class independent noise when $\eta < \frac{k-1}{k}$ (see Theorem 1 of Ghosh et al. (2017)).

---

[1]For this phase we submit the source code as part of supplementary materials to preserve anonymity, however, the final version will contain a link to our GitHub repository.

Moreover, if the population risk of the optimum classifier $\mathbb{E}[\ell(f^*(x,\theta),\hat{y})] = 0$ then $\ell$ is also noise tolerant for class dependent noise. Among the commonly used losses, only the mean absolute error (MAE) satisfies the symmetry constraint. On the other end, the cross-entropy (CE) loss which is widely used for classification, not only does not satisfy the above constraint, but is also an unbounded loss making it extremely susceptible to noise. Specifically, as noise makes the predicted probability of the correct class approach zero, its cross-entropy loss will approach infinity.

Following the work of Ghosh et al. (2017), many studies (Zhang & Sabuncu, 2018; Feng et al., 2020; Wang et al., 2019b; Ma et al., 2020; Wang et al., 2019a) found that models trained with MAE struggle to converge and proposed novel losses that rely on boundedness to achieve robustness instead of the symmetry constraint to avoid convergence problems.

In the following, we use the *weighted mean trick* to bound a loss function to improve its noise robustness. The next lemma specify the requirements for the weights:

**Lemma 5.** *Let $\ell(v,y)$ represent an unbounded loss function and $w(v,y)$ a corresponding non-negative and bounded weight function, then the product $w(v,y)\ell(v,y)$ is bounded if there exists a finite threshold $\mathcal{L}_0$ such that when $\ell(v,y) \geq \mathcal{L}_0$ the corresponding weights $w(v,y)$ are 0.*

*Proof.* Since when $\ell(v,y) \geq \mathcal{L}_0$ the corresponding weights $w(v,y) = 0$ the product $w(v,y)\ell(v,y)$ is also 0. When $\ell(v,y) < \mathcal{L}_0$ since the weights are bounded the product $w(v,y)\ell(v,y)$ is also bounded and thus $0 \leq w(v,y)\ell(v,y) \leq B$ where $B$ is a positive constant that depends on the weight and loss function. $\qquad\square$

The simplest solution that satisfies the above requirement is the negative variance penalization, $\lambda_2 < 0$, as the next lemma shows:

**Lemma 6.** *For weights $w(v,y)$ computed using Theorem 3 with $\lambda_1 > 0, \lambda_2 < 0, \lambda_i = 0, \forall i > 2$ and with negative weights clipped to 0, penalties $\ell(v,y) \geq \mathcal{L}_0$ with $\mathcal{L}_0 = \mathbb{E}[\ell] - \frac{\lambda_1}{\lambda_2}$ will have an associated weight of 0.*

*Proof.* From Theorem 3 the weights are calculated as $w(v,y) = \lambda_1 + \lambda_2(\ell(v,y) - \mathbb{E}[\ell(v,y)])$ from which we can determine the threshold value $\mathcal{L}_0 = \mathbb{E}[\ell] - \frac{\lambda_1}{\lambda_2}$. Thus due to the non-negativity constraint $\ell(v,y) \geq \mathcal{L}_0$ will have an associated weight of 0. $\qquad\square$

Of note, bounding an unbounded loss function is achieved through clipping, however, the resulting loss function is not convex as shown in Figure 3. With parameters $\lambda_1$ and $\lambda_2$ along with $\mathbb{E}[\ell]$ establishing which samples are considered noise and excluded from training. Since for $\lambda_2 < 0$ the threshold $\mathcal{L}_0 = \mathbb{E}[\ell] - \frac{\lambda_1}{\lambda_2}$ is greater than $\mathbb{E}[\ell]$, thus all excluded samples have a loss value above average. Moreover, the higher the magnitude of $\lambda_2$, the closer the threshold $\mathcal{L}_0$ is to $\mathbb{E}[\ell]$. This can be seen in the left plot of Figure 3. Moreover, the magnitude of $\lambda_2$ dictates by how much the loss values below average are amplified and the ones above average suppressed. In practice, using larger values for $\lambda_2$ will decrease the impact of misclassified samples and those near the separating hyperplane on the placement of the decision boundary. Lemma 6 can be extended to include other higher-order moments. In this case, the roots of the weights polynomial will determine which samples participate in the optimization problem and which are considered noise. For example, if the decision boundary should be decided by samples with average loss values then a possible solution is to penalize skewness. Since in this case the weights are defined by a quadratic equation, and using a negative value for $\lambda_3$ will make the parabola open downwards and thus assign non-zero weights only to samples around the mean.

Moreover, we also develop a convex version of the variance penalized cross-entropy by constraining the second derivative of the weighted loss function to be non-negative. The cross-entropy loss is defined as $\ell_{ce}(v,y) = -\log v[y]$ where $v[y]$ represents the predicted probability of sample having the label $y$. To simplify the notation, we will rewrite the cross-entropy loss as $\ell_{ce}(u) = -\log u$ where $u = v[y]$ and similarly rewrite the weight function as $w(u) = \lambda_1 + \lambda_2(\ell(u) - \mathbb{E}[\ell(u)])$. We note that the second derivative of the weighted loss $w(u)\ell(u)$ is 0 for $u = u_c$ with $u_c = \exp\left(1 + \frac{\lambda_1}{2\lambda_2} - \frac{\mathbb{E}[\ell]}{2}\right)$ and is negative for $u < u_c$. To constrain the second derivative be non-negative we linearly interpolate $w(u)\ell(u)$ for $u < u_c$ using the derivative of $w(u)\ell(u)$ at $u_c$. The weight function for $u < u_c$

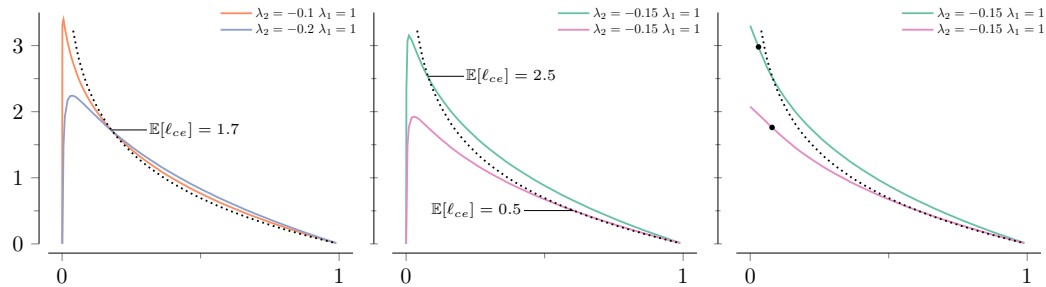

Figure 3: Bounding the cross-entropy loss function using negative variance penalization. Classical cross-entropy function is shown a dotted line. The impact of the $\lambda_2$ parameter on the loss shape when the average loss is constant is shown on the left plot and the impact of the average loss on the shape in the center plot. Right plot shows the convex version of the variance penalized cross-entropy loss.

is $w(u) = \left[ w(u_c)\ell(u_c) + \frac{\mathrm{d}}{\mathrm{d}u}w(u)\ell(u)\big|_{u=u_c}(u - u_c) \right] / \ell(u)$ with the resulting convex objective shown on the right plot of Figure 3. The black dot marks the transition point $u_c$ from the weighted cross-entropy on the right to the linear interpolation on the left of the dot. Of note, the resulting objective is still bounded albeit with a higher upper bound.

In general, methods for training of neural networks in the presence of noise can be classified into two broad categories: noise model-free and noise model-based strategies. Noise model-free methods focus on reducing the impact of outliers and the two main subcategories are: robust losses (Bartlett et al., 2006; Ghosh et al., 2017; Wang et al., 2019a; Zhang & Sabuncu, 2018; Wang et al., 2019b; Natarajan et al., 2013; Mnih & Hinton, 2012; Xu et al., 2019; Patrini et al., 2016; Rooyen et al., 2015; Feng et al., 2020; Ma et al., 2020) and learning management such as meta-learning or regularization (Li et al., 2017; Szegedy et al., 2016; Hendrycks et al., 2019; Liu et al., 2020; Harutyunyan et al., 2020; Lugosi & Mendelson, 2021; Laforgue et al., 2021; Lecué et al., 2020; Li, 2017). On the other hand, noise model-based methods estimate the properties of the noise and use this information when train the model. Noise model-based methods can be further divided into: using noise transition matrix (Patrini et al., 2017; Hendrycks et al., 2018; Chen & Gupta, 2015; Bekker & Goldberger, 2016; Goldberger & Ben-Reuven, 2017; Sukhbaatar et al., 2015; Xia et al., 2019; Yao et al., 2019; Wang et al., 2020; Xia et al., 2020; Yao et al., 2020; Lukasik et al., 2020), noise mitigation requiring a clean dataset (Jiang et al., 2018; Ren et al., 2018; Veit et al., 2017; Zhang et al., 2020; Yuan et al., 2018; Vahdat, 2017; Mirzasoleiman et al., 2020), or incrementally estimating the distribution of clean data (Liu et al., 2017; Zheng et al., 2020; Arazo et al., 2019; Zhang et al., 2017; Ghosh & Lan, 2021; Zhang et al., 2021; Wu et al., 2020).

**Experimental results.** The procedure adopted for all the experiments and elaborated in Appendix B is similar to other studies investigating robust losses (Zhang & Sabuncu, 2018; Feng et al., 2020; Wang et al., 2019b; Ma et al., 2020). The source code to reproduce our results is available online.[1]

The model accuracy for clean data when training with both, class independent and class dependent noise, are summarized in Table 1. Of note, we report the accuracy results of the model at the end of the training to capture the overfitting on noise for some models. We observe that the training with moments penalization outperforms other methods in low to moderate noise level scenarios and falls behind in high noise scenarios but still above the classical cross-entropy. The performances of the convex version and non-convex version are very similar, however, the convex version shows better results in high noise scenarios under class dependent noise. Additional results when penalizing higher-order moments are provided in Appendix C.

In case of CIFAR datasets, for class independent noise and for $\eta = 0.2$ the proposed method improves the accuracy of the classical cross-entropy by $6\%$, the highest improvement among the investigated methods. However, this improvement decreases as the noise ratio increases and approaches the accuracy of the classical cross entropy loss for $\eta = 0.8$. To note, the convex version of the moments method for $\eta = 0.8$ underpeformed by 0.8% compared to the classical cross-entropy, however, in

---

[1]For this phase we submit the source code as part of supplementary materials to preserve anonymity, however, the final version will contain a link to our GitHub repository.

Table 1: Mean accuracy (%) and standard deviation on clean data over 5 runs. The best result for each scenario is underlined.

| | Loss | Class independent | | | | Class dependent | | | |
|---|---|---|---|---|---|---|---|---|---|
| | | 0.2 | 0.4 | 0.6 | 0.8 | 0.1 | 0.2 | 0.3 | 0.4 |
| CIFAR-10 | Classical | $84.1_{0.2}$ | $77.1_{0.4}$ | $66.3_{0.4}$ | $36.0_{1.5}$ | $88.9_{0.2}$ | $87.2_{0.2}$ | $84.8_{0.4}$ | $\underline{80.9}_{0.7}$ |
| | Moments | $\underline{90.5}_{0.1}$ | $85.0_{0.6}$ | $69.1_{0.4}$ | $37.3_{2.3}$ | $\underline{91.4}_{0.2}$ | $\underline{89.9}_{0.0}$ | $85.0_{3.2}$ | $72.9_{3.3}$ |
| | Moment-convex | $86.8_{0.6}$ | $80.3_{0.3}$ | $67.6_{0.6}$ | $35.2_{1.5}$ | $89.1_{0.1}$ | $87.2_{0.2}$ | $83.5_{0.3}$ | $78.7_{0.4}$ |
| | TERM | $89.5_{0.1}$ | $83.2_{0.2}$ | $69.2_{0.5}$ | $37.2_{2.0}$ | $91.1_{0.2}$ | $88.1_{0.4}$ | $83.0_{0.7}$ | $77.2_{1.1}$ |
| | Taylor | $90.2_{0.2}$ | $86.2_{0.1}$ | $73.0_{0.5}$ | $10.0_{0.0}$ | $91.0_{0.2}$ | $88.4_{0.3}$ | $83.7_{0.4}$ | $78.0_{1.1}$ |
| | Normalized | $90.1_{0.1}$ | $\underline{87.0}_{0.2}$ | $\underline{80.8}_{0.2}$ | $32.6_{2.9}$ | $90.9_{0.1}$ | $89.7_{0.3}$ | $\underline{86.8}_{0.5}$ | $79.8_{0.5}$ |
| | Symmetric | $90.0_{0.2}$ | $86.7_{0.2}$ | $80.2_{1.0}$ | $\underline{44.8}_{4.0}$ | $90.8_{0.2}$ | $89.6_{0.1}$ | $86.5_{0.4}$ | $79.8_{0.9}$ |
| | Generalized | $90.4_{0.2}$ | $86.0_{0.2}$ | $67.3_{2.7}$ | $10.0_{0.0}$ | $91.3_{0.1}$ | $89.0_{0.1}$ | $83.7_{0.5}$ | $74.0_{3.2}$ |
| CIFAR-100 | Classical | $40.0_{0.5}$ | $31.1_{1.1}$ | $20.7_{0.5}$ | $11.6_{0.4}$ | $46.8_{0.2}$ | $42.6_{0.4}$ | $37.2_{0.5}$ | $31.7_{0.3}$ |
| | Moments | $\underline{46.1}_{0.4}$ | $39.8_{0.7}$ | $27.9_{0.8}$ | $13.8_{1.0}$ | $\underline{47.8}_{0.9}$ | $43.7_{0.9}$ | $37.2_{0.4}$ | $30.6_{0.3}$ |
| | Moment-convex | $44.5_{0.8}$ | $36.7_{0.9}$ | $24.8_{0.8}$ | $13.4_{0.8}$ | $47.4_{0.5}$ | $42.9_{0.7}$ | $36.9_{0.5}$ | $31.1_{0.6}$ |
| | TERM | $45.5_{0.3}$ | $\underline{41.5}_{1.1}$ | $\underline{32.8}_{1.4}$ | $17.7_{0.4}$ | $47.1_{0.6}$ | $\underline{44.6}_{0.7}$ | $37.9_{0.6}$ | $30.9_{0.6}$ |
| | Taylor | $37.3_{0.6}$ | $33.0_{1.3}$ | $27.2_{0.8}$ | $17.4_{0.2}$ | $38.7_{0.6}$ | $35.9_{0.7}$ | $31.0_{1.0}$ | $26.6_{0.5}$ |
| | Normalized | $32.9_{0.8}$ | $27.8_{0.7}$ | $22.7_{0.4}$ | $13.9_{0.2}$ | $35.0_{0.3}$ | $32.7_{0.3}$ | $30.2_{0.4}$ | $26.5_{0.6}$ |
| | Symmetric | $43.1_{0.4}$ | $37.9_{0.4}$ | $31.2_{0.8}$ | $\underline{19.4}_{0.5}$ | $45.3_{0.5}$ | $43.1_{0.2}$ | $\underline{41.0}_{0.5}$ | $\underline{34.6}_{0.3}$ |
| | Generalized | $38.2_{0.4}$ | $33.7_{0.9}$ | $28.1_{0.6}$ | $18.5_{0.4}$ | $39.6_{0.9}$ | $37.9_{0.7}$ | $35.8_{0.6}$ | $29.9_{0.3}$ |
| Fashion-MNIST | Classical | $92.0_{0.1}$ | $90.9_{0.1}$ | $89.0_{0.2}$ | $78.4_{1.0}$ | $\underline{92.9}_{0.1}$ | $\underline{92.5}_{0.3}$ | $92.0_{0.7}$ | $90.6_{1.7}$ |
| | Moments | $92.1_{0.2}$ | $91.6_{0.2}$ | $89.8_{0.1}$ | $80.2_{1.1}$ | $87.0_{2.9}$ | $82.4_{6.7}$ | $79.3_{6.8}$ | $62.1_{3.0}$ |
| | Moments-convex | $\underline{92.4}_{0.1}$ | $91.6_{0.1}$ | $89.6_{0.2}$ | $79.7_{1.0}$ | $92.6_{0.2}$ | $92.0_{0.4}$ | $\underline{92.4}_{0.2}$ | $88.7_{1.5}$ |
| | TERM | $92.1_{0.1}$ | $91.6_{0.1}$ | $90.0_{0.1}$ | $80.3_{0.9}$ | $92.3_{0.1}$ | $92.0_{0.4}$ | $91.5_{0.9}$ | $91.3_{0.4}$ |
| | Taylor | $90.6_{0.0}$ | $89.4_{0.3}$ | $86.8_{0.2}$ | $75.7_{0.7}$ | $91.1_{0.1}$ | $90.7_{0.4}$ | $89.8_{1.4}$ | $85.1_{3.8}$ |
| | Normalized | $91.5_{0.1}$ | $90.8_{0.3}$ | $88.9_{0.3}$ | $79.8_{1.3}$ | $91.7_{0.1}$ | $91.7_{0.3}$ | $90.7_{0.7}$ | $91.1_{0.1}$ |
| | Symmetric | $92.3_{0.1}$ | $\underline{91.8}_{0.1}$ | $\underline{90.6}_{0.3}$ | $\underline{82.8}_{1.2}$ | $92.5_{0.1}$ | $92.3_{0.3}$ | $91.9_{0.6}$ | $\underline{91.5}_{1.3}$ |
| | Generalized | $91.2_{0.2}$ | $90.2_{0.2}$ | $88.1_{0.2}$ | $73.9_{1.1}$ | $91.6_{0.1}$ | $91.3_{0.3}$ | $90.7_{0.7}$ | $90.4_{0.3}$ |

all other scenarios it had a higher accuracy. High noise scenarios turned out to be challenging for all losses with the Taylor and Generalized cross-entropy losses not converging on CIFAR-10 for $\eta = 0.8$ and the best accuracy was registered by Symmetric cross-entropy loss proposed by Wang et al. (2019b). For class dependent noise, the moments penalization registered the highest accuracy for CIFAR-10 when $\eta = 0.1$ and $\eta = 0.2$ but had the lowest accuracy for $\eta = 0.4$. To further investigate this behavior, we reran the experiments and monitored the accuracy for each individual class. The classical cross-entropy recorded similar accuracy as the robust methods on classes affected by noise and outperformed them on classes 6 and 8, both not targeted by class dependent noise. The poor performance of robust methods can be justified by using parameters tuned for low and moderate noise. In case of Fashion-MNIST dataset, as similar behavior as with CIFAR datasets was observed.

## 5 CONCLUSION

The main goal of the current work was to investigate the optimization of a weighted mean and the flexibility it provides in enforcing properties such as robustness when training with noisy labels. In addition, we extended previous variance penalization methods to include higher-order moments while eliminating some of their limitations. One of the significant findings to emerge from this study was that we can control the final distribution of the loss values by penalizing higher-order moments. In particular, by enforcing the distribution of the cross-entropy to have a higher variance through negative variance penalization, we improved the models' accuracy when trained with noisy labels. Although this paper centers on classification problems, the framework can also control the bias-variance trade-off and can also be applicable to regression problems.

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

# A  PROOFS AND ADDITIONAL THEORETICAL RESULTS

First, we present a proposition used in the proof of Theorem 1.

**Proposition 7.** *Let $U$ and $V$ be two random variables, and let $\overline{U}$ and $\overline{V}$ denote their expectation, then:*

$$\mathbb{E}\left[\left(U - \overline{U}\right)V\right] = \mathbb{E}\left[\left(U - \overline{U}\right)\left(V - \overline{V}\right)\right] \tag{7}$$

*Proof.* To prove we write $V$ as $\left(V - \overline{V}\right) + \overline{V}$ and then use the linear property of the expectation operator:

$$\begin{aligned}
\mathbb{E}\left[\left(U - \overline{U}\right)V\right] &= \mathbb{E}\left[\left(U - \overline{U}\right)\left(\left(V - \overline{V}\right) + \overline{V}\right)\right] \\
&= \mathbb{E}\left[\left(U - \overline{U}\right)\left(V - \overline{V}\right)\right] + \mathbb{E}\left[\left(U - \overline{U}\right)\overline{V}\right] \\
&= \mathbb{E}\left[\left(U - \overline{U}\right)\left(V - \overline{V}\right)\right] + \overline{V}\left(\mathbb{E}\left[U\right] - \overline{U}\right) \\
&= \mathbb{E}\left[\left(U - \overline{U}\right)\left(V - \overline{V}\right)\right]
\end{aligned}$$

$\square$

Next we present two propositions used in the proofs of the main theorems of this paper.

**Proposition 8.** *Let $\ell : \mathbb{R}^n \to [0, \infty)$ and $p : \mathbb{R} \to \mathbb{R}$ two convex functions, then the composition $w(x) = p(\ell(x) - M)$ where $M \geq 0$ is convex if $p$ is non decreasing.*

*Proof.* Using the convexity of $\ell$ for any $x, y \in \mathbb{R}^n$ and $\alpha \in [0, 1]$ we have:

$$\ell(\alpha x + (1 - \alpha)y) \leq \alpha\ell(x) + (1 - \alpha)\ell(y) \tag{8}$$

Using the fact that $p$ is non decreasing along we the above inequality we obtain:

$$p(\ell(\alpha x + (1 - \alpha)y)) \leq p(\alpha\ell(x) + (1 - \alpha)\ell(y)) \tag{9}$$

and using convexity of $p$ for the term on the right we get:

$$p(\alpha\ell(x) + (1 - \alpha)\ell(y)) \leq \alpha p(\ell(x)) + (1 - \alpha)p(\ell(y)) \tag{10}$$

Combining the two inequalities leads to the following result $(\ell(\alpha x + (1 - \alpha)y)) \leq \alpha p(\ell(x)) + (1 - \alpha)p(\ell(y))$ which proves the composition $p(\ell(x))$ is convex. Moreover, since $p$ is convex and $\ell(x) - M$ is in the domain of $p$ then $p(\ell(x) - M)$ is also convex as it is a composition with an affine mapping. This proves that $w(x)$ is convex.

$\square$

**Proposition 9.** *Let $f, g : \mathbb{R}^n \to [0, \infty)$ be two convex function taking non-negative values then their product $h(x) = f(x)g(x)$ is convex if $[f(x) - f(y)][g(x) - g(y)] \geq 0, \forall x, y \in \mathbb{R}^n$.*

*Proof.* The function $h$ is convex if it satisfies the following inequality where $\alpha \in [0, 1]$:

$$0 \leq \alpha h(x) + (1 - \alpha)h(y) - h(\alpha x + (1 - \alpha)y) \tag{11}$$
$$0 \leq \alpha(fg)(x) + (1 - \alpha)(fg)(y) - (fg)(\alpha x + (1 - \alpha)y) \tag{12}$$

Note that since the two functions are convex the following inequalities hold:

$$f(\alpha x + (1 - \alpha)y) \leq \alpha f(x) + (1 - \alpha)f(y) \tag{13}$$
$$g(\alpha x + (1 - \alpha)y) \leq \alpha g(x) + (1 - \alpha)g(y) \tag{14}$$

Given that the two functions take non-negative values we can multiply the above two inequalities.

$$f(\alpha x + (1-\alpha)y)g(\alpha x + (1-\alpha)y) \leq [\alpha f(x) + (1-\alpha)f(y)] [g(x) + (1-\alpha)g(y)] \quad (15)$$
$$(fg)(\alpha x + (1-\alpha)y) \leq [\alpha f(x) + (1-\alpha)f(y)] [g(x) + (1-\alpha)g(y)] \quad (16)$$

Substituting $(fg)(\alpha x + (1-\alpha)y)$ from 16 in inequality 11 we obtain:

$$0 \leq \alpha(fg)(x) + (1-\alpha)(fg)(y) - [\alpha f(x) + (1-\alpha)f(y)] [g(x) + (1-\alpha)g(y)] \quad (17)$$

Multiplying the two square brackets and then grouping the terms we get:

$$0 \leq \alpha(1-\alpha) [f(x) - f(y)] [g(x) - g(y)] \quad (18)$$

We complete the proof by noting that $\alpha \geq 0$ and thus the product of two convex functions is convex if $[f(x) - f(y)] [g(x) - g(y)] \geq 0$. $\qquad \square$

**Lemma 2.** *As introduced in Theorem 1, the variance expansion of a convex loss function $\ell(v, y)$ in $v$ yields a new objective that is also convex in $v$ if $\lambda \in [0, \lambda_{max}]$, where $\lambda_{max} = 1/(\mathbb{E}[\ell] - \min \ell)$.*

*Proof.* Since $y$ are constants and take finite values we can index both, loss and weight functions, using $y$ as $\ell_y(v) = \ell(v, y)$ and $w_y(v) = w(v, y)$. We note that $w_y(v) = p(\ell_y(v) - \mathbb{E}[\ell])$ where $p$ is a polynomial of degree 1 with $p(t) = \lambda t + 1$. First we use the result of Proposition 8 and get that $w_y(v)$ is convex if $\lambda$ is non negative thus the lower bound of the interval. Next, we use Proposition 9 which states that $w_y(v)\ell_y(v)$ is convex if $[w_y(v) - w_y(u)] [\ell_y(v) - \ell_y(u)] \geq 0$ and $w_y(v)$ is non negative. Since $p$ is a linear function we can use the equality $p(a) - p(b) = \lambda(a - b)$, with $a = \ell_y(v) - \mathbb{E}[\ell]$ and $b = \ell_y(u) - \mathbb{E}[\ell]$ and replace $w_y$ in the previous inequality to obtain $\lambda[\ell(x) - \ell(y)]^2 \geq 0$ which is always true for $\lambda > 0$. However, $w_y(v)$ is non negative only for $\lambda \leq 1/(\min \ell - \mathbb{E}[\ell])$ which proves the upper bound of the interval. $\qquad \square$

**Theorem 3** (Moments Expansion). *Let $\ell$ be a loss function with finite first $m$ central moments and define $w(v, y) = \sum_{i=1}^{m} \lambda_i(\ell(v, y) - \mathbb{E}[\ell(v, y)])^{i-1}$, then we have:*

$$\mathbb{E}[w\ell] = \lambda_1 \mathbb{E}[\ell] + \sum_{i=2}^{m} \tilde{\lambda}_i \mathbb{E}\left[(\ell - \mathbb{E}[\ell])^i\right] \quad (4)$$

*where $\tilde{\lambda}_i = \lambda_i + \lambda_{i+1}\mathbb{E}[\ell]$ for $i < m$ and $\tilde{\lambda}_m = \lambda_m$*

*Proof.* In this case, for $i \geq 2$ we cannot apply Lemma 7 since the expression $(\ell - \mathbb{E}[\ell])^i$ is not guaranteed to have zero mean and as a result it incurs an additional penalization of the previous moment. The proof follows the same steps as Theorem 1:

$$\mathbb{E}[W\ell] = \mathbb{E}\left[\sum_{i=1}^{m} \lambda_i(\ell - \mathbb{E}[\ell])^{i-1}\ell\right]$$

$$= \sum_{i=1}^{m} \lambda_i \mathbb{E}\left[(\ell - \mathbb{E}[\ell])^{i-1}\ell\right]$$

$$= \lambda_1 \mathbb{E}[\ell] + \sum_{i=2}^{m} \lambda_i \mathbb{E}\left[(\ell - \mathbb{E}[\ell])^{i-1}\ell\right]$$

$$= \lambda_1 \mathbb{E}[\ell] + \sum_{i=2}^{m} \lambda_i \mathbb{E}\left[(\ell - \mathbb{E}[\ell])^{i-1}(\ell - \mathbb{E}[\ell] + \mathbb{E}[\ell])\right]$$

$$= \lambda_1 \mathbb{E}[\ell] + \sum_{i=2}^{m} \lambda_i \mathbb{E}\left[(\ell - \mathbb{E}[\ell])^i\right] + \sum_{i=2}^{m} \lambda_i \mathbb{E}[\ell]\mathbb{E}\left[(\ell - \mathbb{E}[\ell])^{i-1}\right]$$

$$= \lambda_1 \mathbb{E}[\ell] + \sum_{i=1}^{m-1} \tilde{\lambda}_i \mathbb{E}\left[(\ell - \mathbb{E}[\ell])^i\right] + \lambda_m \mathbb{E}\left[(\ell - \mathbb{E}[\ell])^m\right]$$

For the last step, we combine the two sums by matching the $\mathbb{E}\left[(\ell - \mathbb{E}[\ell])^i\right]$ terms and consolidate the penalization factors as $\tilde{\lambda}_i = \lambda_i + \lambda_{i+1}\mathbb{E}[\ell]$. $\qquad\square$

**Lemma 4** (Convexity of Moments Expansion). *Let $\ell(v, y)$ be a loss function convex in $v$ and $p : \mathbb{R} \to [0, \infty)$ be a non negative and differentiable convex function and $M \geq 0$, then the weighted objective $w(v, y)\ell(v, y)$ with $w(v, y) = p(\ell(v, y) - M)$ is convex in $v$ if $p$ is non decreasing.*

*Proof.* Since $y$ are constants and take finite values we can index both, loss and weight functions, using $y$ such as $\ell_y(v) = \ell(v, y)$ and $w_y(v) = w(v, y)$. With $w_y(v) = p(\ell_y(v) - M)$ and using the result of Proposition 8 we obtain that $w_y(v)$ is convex as $p$ is non decreasing. Next, from Proposition 9 and since $p$ takes non negative values we get that the product $w_y(v)\ell_y(v)$ is convex if $[w_y(v) - w_y(u)][\ell_y(v) - \ell_y(u)] \geq 0$. Since $p$ is convex and differentiable we use the first order condition $p(a) - p(b) \geq p'(b)(a - b)$ with $a = \ell_y(v) - M$ and $b = \ell_y(u) - M$ to obtain:

$$p(\ell_y(v) - M) - p(\ell_y(u) - M) \geq p'(\ell_y(u) - M)[\ell_y(v) - M - \ell_y(u) + M] \qquad (19)$$

$$w_y(v) - w_y(u) \geq p'(\ell_y(u) - M)[\ell_y(v) - \ell_y(u)] \qquad (20)$$

Substituting this result in the requirement from Proposition 9 we obtain:

$$p'(\ell_y(v) - M)[\ell_y(v) - \ell_y(u)]^2 \geq 0 \qquad (21)$$

Given that $p$ is non decreasing implies $p'(\ell(y) - M) \geq 0$ and proves that the inequality always holds. $\qquad\square$

## B  Experimental Details

After we corrupt the training datasets (CIFAR-10, CIFAR-100, Fashion-MNIST) with noise we retain 10% as a secondary validation dataset. This allows us to detect when the model is overfitting on noise by comparing the performance on clean versus noisy validation data. For the class independent noise, we flip the label to any other class with equal probability such that the ratio of noisy labels is $\eta$. For class dependent noise, for CIFAR-10 and Fashion-MNIST we only flip the label in the source classes $\{9, 2, 3, 5, 4\}$ to the corresponding target class $\{1, 0, 5, 3, 7\}$ given the noise ratio $\eta$. And for CIFAR-100 we flip between two randomly selected subclasses withing each superclass. We use the same modes as in Wang et al. (2019b) when training on CIFAR-10 which is an 8 layer network composed of 6 convolutional layers followed by 2 fully connected layers. For Fashion-MNIST the model contains 4 convolutional layers followed by 3 fully connected layers. For CIFAR-100 the model we use is ResNet-34 (He et al., 2016). We train using SGD with 0.9 momentum, 0.005 weight decay for the convolutional layers and 0.01 for the fully connected layers, and a starting learning rate of 0.01 which we divide by 10 every 20 epochs for a total of 60 training epochs for CIFAR datasets and every 5 epochs for a total of 15 training epochs for Fashion-MNIST. The training batch size is 128 samples.

We compare our solution for robustification of the loss function through moments penalization against five other state of the art methods: $i)$ Tilted empirical risk minimization (TERM) introduced by Li et al. (2021), $ii)$ Taylor expansion of cross entropy (Taylor) proposed by Feng et al. (2020), $iii)$ Normalized cross entropy coupled with reverse cross entropy (Normalized) investigated by Ma et al. (2020), $iv)$ Symmetric cross entropy (Symmetric) studied by Wang et al. (2019b), and $v)$ Generalized cross entropy (Generalized) explored by Zhang & Sabuncu (2018).

Zhang & Sabuncu (2018) proposed a generalized cross-entropy loss parameterized by $q$ that recovers CE loss for $q \to 0$ and MAE for $q = 1$. For intermediate values of $q$, the loss is bounded and trades robustness for convergence. Feng et al. (2020) found that the first two terms of the Taylor series expansion for the CE loss are the MAE and MSE, respectively. And suggested adjusting the number of terms in the Taylor expansion for the CE to balance noise robustness and convergence. Wang et al. (2019b) proposed using a two term loss, complimenting the CE loss with a secondary reverse cross-entropy term that satisfies the symmetry constraint. The research by Ma et al. (2020) extended the concept of two term loss combining a robust "active" loss and a robust "passive" loss.

For these methods, when possible we used the same parameters suggested by their authors, however, when the method underperfomed we used cross-validation to find better ones. When training on CIFAR-10 and Fashion-MNIST datasets we used: $\lambda_1 = 1$, $\lambda_2 = -0.5$ for Moments and its convex version, $t = -0.5$ for TERM, $t = 2$ for Taylor, $\alpha = 10$, $\beta = 1$ for Normalized, $\alpha = 0.1$, $\beta = 1$ for Symmetric, and $q = 0.7$ for Generalized. When training on CIFAR-100 the parameters where: $\lambda_1 = 1$, $\lambda_2 = -0.5$ for Moments and its convex version, $t = -0.5$ for TERM, $t = 6$ for Taylor, $\alpha = 10$, $\beta = 0.1$ for Normalized, $\alpha = 6.0$, $\beta = 0.1$ for Symmetric, and $q = 0.7$ for Generalized.

## C  ADDITIONAL EXPERIMENTAL RESULTS

Table 2 shows the results when penalizing the third (skewness) and fourth (kurtosis) central moments. The resulting polynomial and the corresponding weighted cross-entropy loss are illustrated in Figure 4. When penalizing the third central moment samples with loss values close to the mean receive most weight and the one deviating most from the mean receive the least weight. Experimental results on CIFAR-10 dataset show that when increasing the magnitude of $\lambda_3$ the accuracy for class independent noise rates $\eta = 0.2$ and $\eta = 0.4$ improves by around 1%. Moreover, in both cases the resulting accuracy is higher than that of the classical CE. Similarly, for class dependent noise, the accuracy increased by around 1% when penalizing with $\lambda_3 = -0.5$ for noise rates $\eta = 0.2$ and $\eta = 0.3$. Likewise, when increasing the magnitude of $\lambda_4$ the accuracy for class independent noise rates $\eta = 0.2$ and $\eta = 0.4$ improves by up to 3.5%. However, for class dependent noise the results are similar.

Table 2: Mean accuracy (%) and standard deviation on clean data over 5 runs. The best result for each scenario is underlined.

| | Loss | Noise rate $\eta$ | | | | | | | |
| --- | --- | --- | --- | --- | --- | --- | --- | --- | --- |
| | | Class independent | | | | Class dependent | | | |
| | | 0.2 | 0.4 | 0.6 | 0.8 | 0.1 | 0.2 | 0.3 | 0.4 |
| CIFAR-10 | Classical | 84.1 0.2 | 77.1 0.4 | 66.3 0.4 | 36.0 1.5 | 88.9 0.2 | 87.2 0.2 | 84.8 0.4 | 80.9 0.7 |
| | $\lambda_1 = 1, \lambda_3 = -0.1$ | 87.7 0.1 | 78.8 0.3 | 67.4 1.0 | 35.7 2.1 | 89.6 0.3 | 87.3 0.2 | 84.6 0.1 | 80.6 0.4 |
| | $\lambda_1 = 1, \lambda_3 = -0.5$ | 88.4 0.1 | 80.3 0.2 | 67.4 0.3 | 35.4 2.1 | 89.3 0.2 | 88.3 0.4 | 85.9 0.4 | 76.9 3.0 |
| | $\lambda_1 = 1, \lambda_4 = -0.05$ | 88.4 0.1 | 78.5 0.2 | 67.6 0.8 | 35.6 2.1 | 89.7 0.1 | 87.3 0.2 | 84.9 0.4 | 80.9 0.5 |
| | $\lambda_1 = 1, \lambda_4 = -0.1$ | 88.7 0.2 | 82.0 0.3 | 66.8 0.7 | 35.2 2.2 | 89.6 0.4 | 88.0 0.1 | 84.3 0.4 | 80.0 0.6 |

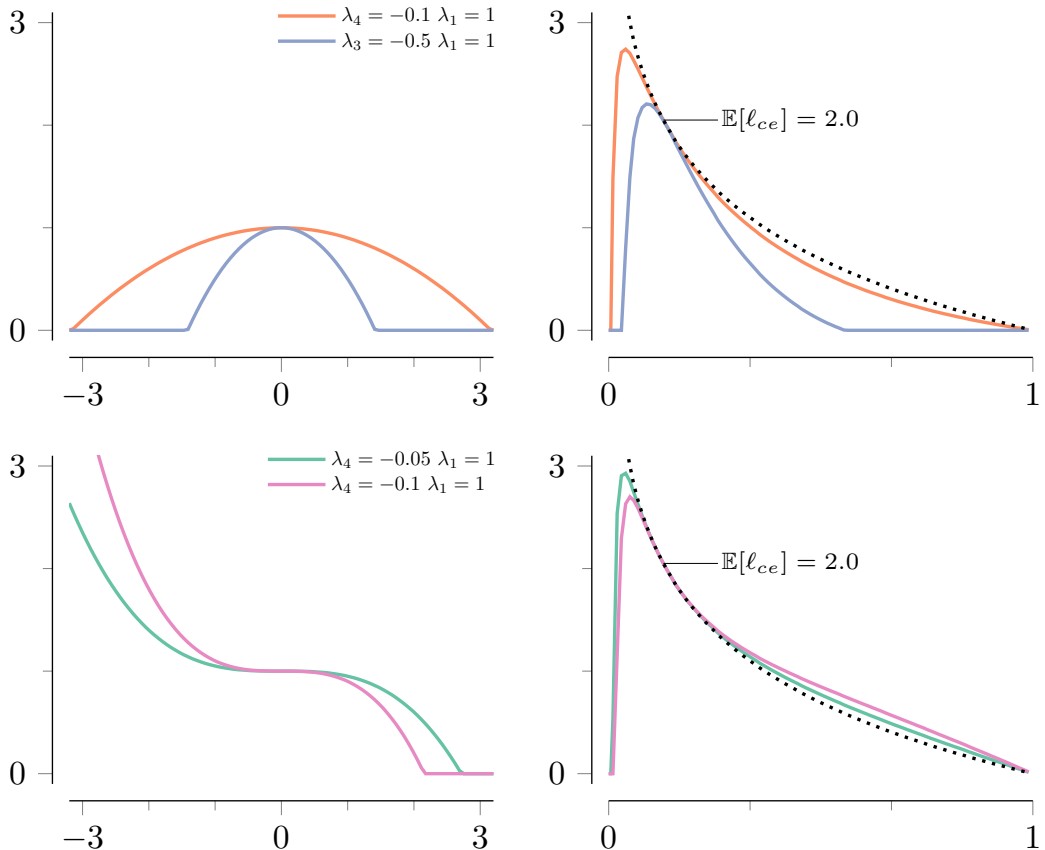

Figure 4: Polynomial functions for moments penalization and the corresponding weighted cross-entropy loss. Left column shows the polynomial functions used for penalizing the third central moment, top, and forth central moment, bottom. Right column shows the weighted cross-entropy function for $\mathbb{E}[\ell] = 2$. Dotted line shows the classical cross entropy for reference.

