# OpenReview forum: "The weighted mean trick – optimization strategies for robustness"
_ICLR.cc/2022/Conference — ICLR 2022 Submitted_

### Official Review · Reviewer_fmnG · 2021-10-26

**Correctness:** 4
**Technical Novelty And Significance:** 3
**Empirical Novelty And Significance:** 3
**Recommendation:** 6
**Confidence:** 4

**Main Review:**

Strengths:
I think this is an interesting observation and could potentially have consequences in several robust learning algorithms.

Weaknesses:
I would have liked to see some applications to existing results that optimize the weights to solve estimation problems in outlier-robust settings -- for instance [1]. But there are several other works that also do such things.

[1] "Robust Sparse Estimation Tasks in High Dimensions" Li'17

**Summary Of The Paper:**

In this paper, the authors demonstrate that minimizing the weighted mean results in higher oder moments of the loss distribution.

They do this by explicitly demonstrating specific choices for the weights such that the expected weighted loss actually corresponds to minimizing the original loss regularized by a linear combination of the higher moments. They also note the ranges of the regularization parameter that preserve convexity.

The authors state their work in the context of two recent papers [1], [2] which attempt. to control the bias-variance tradeoff in different ways, effectively controlling either just the variance, or all the higher moments. In contrast, this technique allows them to control any specific combination of higher moments.

[1] "Variance-based regularization with convex objectives" Duchi, Namkoong'19

[2] "Tilted Empirical Risk Minimization" Li, Beirami, Sanjabi, Smith'20

**Summary Of The Review:**

I think this is an interesting paper and vote to accept it. I think the observation carries value and might have several applications down the line.

The only detracting aspect is that the theorems are relatively straightforward, and so I would have liked to see more consequences of this observation in the context of other algorithms that exist in the literature.

---

> ### Author Response · Authors · 2021-11-25
>
> We thank the reviewers for the feedback. To access the pdfdiff between the initial submission and the most updated version, please click the “Show Revisions” link between the title and abstract (requires login) followed by the “Compare Revisions” button and select the first (imported: 22 Nov 2021) and third (modified: 05 Oct 2021) entries and click on the “View Differences” button located on the top right to see the changes highlighted.
>
> The main changes of the paper are as follows:
> - We reported new experimental results of penalizing higher-order moments.
> - We included a new section entitled “Notations” containing the conventions and notations.
> - We updated the notations used in the theoretical results concerning the moments penalization and robustness.
>
> In response to “Weaknesses: I would have liked to see some applications to existing results that optimize the weights to solve estimation problems in outlier-robust settings -- for instance [1]. But there are several other works that also do such things.”
>
> We thank the reviewer for the suggested paper, we studied and cited it.
>
> To answer how the suggested approach can be related to the current work, we see an interesting research direction of computing the weights using the empirical class mean instead of the empirical population mean. Given that there are fewer samples to calculate the class mean which are also contaminated with noise, a robust multivariate mean estimation method would be required such as the one proposed in Li'17.
>
> Unfortunately, we could not complete such an in-depth investigation within the two-week time period. However, this research direction sparked our interest and we will investigate it in the future.

---

### Official Review · Reviewer_Mbr3 · 2021-11-02

**Correctness:** 3
**Technical Novelty And Significance:** 2
**Empirical Novelty And Significance:** 2
**Recommendation:** 5
**Confidence:** 3

**Main Review:**

Here are my comments on the paper:

(1) Theorem 1 assumes that the first moment of $\ell$ exists. What if $\ell$ has heavy-tailed distribution and its mean may not even exist?

(2) The statement of Lemma 2 is quite weird. I think $\ell$ is a random variable. What do the authors mean by "convex objective $\ell$"? What does $\min(\ell)$ mean? Does that notation stand for the minimum coordinate of $\ell$?

(3) " When using the weighted mean trick to penalize the variance and clip the negative weights to 0 then the objective remains convex for any positive": Why is it true?

(4) When using the weighted mean trick, what is the concentration behavior of the samples from $W \ell$ around their expectation $E(W \ell)$? I also have a similar question for moments expansion in Theorem 3.

(5) When the weights are clipped to zero due to their negative values, will the expansions in Theorems 1 and 3 still hold?

(6) "However, for moderately small penalization factors the objective remains convex on almost the entirety of the domain and does not hinder the convergence": I am not so sure about how to quantify "moderately small". The authors need to explain this point further.

(7) "In what follows, we show that using a negative variance penalization factor,  $\lambda_{2} < 0$, bounds the loss function": I wonder the relation of $\lambda_{2}$ and $\eta$, which is class label misspecification probability. Intuitively, $\lambda_{2}$ should depend on $\eta$.

(8) I do not understand the writing in Page 7. Several parts are written without proper explanation. The authors may consider rewriting the writing in Page 7 to improve the readability of the paper.

**Summary Of The Paper:**

In the paper, the authors proposed minimizing a simple weighted mean, which leads to an optimization problem of the high-order moments of the loss distribution. Experiment results show the weighted mean trick has similar performance as other robust loss functions.

**Summary Of The Review:**

In my opinion, the contribution of the paper is quite marginal and the paper is quite hard to understand, which causes by the poor writing and lacks of proper explanation.

---

> ### Author Response · Authors · 2021-11-25
> **We thank the reviewer for the feedback**
>
> We thank the reviewers for the detailed feedback and constructive comments. We followed the reviewer’s suggestions closely and updated the paper. We hope the changes to the revised manuscript address all the issues. To access the pdfdiff between the initial submission and the most updated version, please click the “Show Revisions” link between the title and abstract (requires login) followed by the “Compare Revisions” button and select the first (imported: 22 Nov 2021) and third (modified: 05 Oct 2021) entries and click on the “View Differences” button located on the top right to see the changes highlighted.
>
> We will first summarize the changes and then provide point-by-point answers to all the comments. The main changes are as follows:
> - We included a new section entitled “Notations” containing the conventions and notations.
> - We updated the notations used in the theoretical results concerning the moments penalization and robustness.
> - We reported new experimental results of penalizing higher-order moments.
>
> Point by point answers:
>
> - In response to “(1) Theorem 1 assumes that the first moment of exists. What if has heavy-tailed distribution and its mean may not even exist?”
>
> The classical ERM will fail in this case and should be replaced with a robust counterpart. As a result, the weighted mean must also be robustified which we think this is an interesting research direction. However, in the current work, we limited our research to the contamination model presented at the beginning of section 4. Robust Classification.
>
> - In response to “(2) The statement of Lemma 2 is quite weird. I think $\ell$ is a random variable. What do the authors mean by "convex objective $\ell$"? What does $min(\ell)$ mean? Does that notation stand for the minimum coordinate of $\ell$?
>
> We hope the new section on notation along with the revised version of Lemma 2 clarifies this confusion. The purpose of Lemma 2 is to show that if negative weights are not clipped to 0, only a limited amount of variance penalization will be possible. Otherwise, convexity will not be preserved.
>
> - In response to “(3) " When using the weighted mean trick to penalize the variance and clip the negative weights to 0 then the objective remains convex for any positive": Why is it true?”
>
> For variance penalization, the weights are computed using a polynomial of degree 1 in $\ell - \mathbb{E}[\ell]$. For the weights function (composition of a polynomial and loss function) to be convex, $\lambda$ must be positive. For the product of weight and loss functions to be convex, both functions must be convex and non-negative. Thus, negative weights must be clipped to 0 to satisfy the non-negativity constraint in order to preserve convexity.
>
> - In response to “(4) When using the weighted mean trick, what is the concentration behavior of the samples from $w\ell$ around their expectation $\mathbb{E}[w\ell]$? I also have a similar question for moments expansion in Theorem 3.”
>
> When the weights are positive the weighted mean is equivalent to the sum of moments. However, when the weights are clipped, this is no longer the case. We think this is an interesting research direction which we will investigate in the future.
>
> - In response to “(5) When the weights are clipped to zero due to their negative values, will the expansions in Theorems 1 and 3 still hold?”
>
> Though the effective penalization will be lower, the expansions do hold. The paper states: “samples with a corresponding zero weight reach their maximum contribution in the moments penalization. Further increasing the penalization factors $\lambda_i$, will have no effect on those samples and only samples with non-zero weights will participate in the training. As a result, the efficiency of the moments penalization will slightly fall.”
>
> - In response to “(6) I am not so sure about how to quantify "moderately small". The authors need to explain this point further.”
>
> We updated the text to refer the reader to Figure 3 where we plot the cross-entropy loss function for multiple values of $\lambda_2$ and $\mathbb{E}[\ell]$.
>
> - In response to “(7) "In what follows, we show that using a negative variance penalization factor, $\lambda_2 < 0$, bounds the loss function": I wonder the relation of $\lambda_2$ and $\eta$, which is class label misspecification probability. Intuitively, $\lambda_2$ should depend on $\eta$.”
>
> For the cross-entropy, a lower upper bound prevents large loss values to skew the mean which usually happens when samples are contaminated with noise. However, the lower upper bound weakens one of the cross-entropy strengths of putting more weight on harder samples. Therefore, one should balance the two effects depending on the training data and the model used.
>
> - In response to “(8) I do not understand the writing in Page 7. Several parts are written without proper explanation.”
>
> We updated the notation and clarified some aspects detailed on page 7. We hope the new revision is more readable.

---

### Official Review · Reviewer_ndU8 · 2021-11-03

**Correctness:** 2
**Technical Novelty And Significance:** 1
**Empirical Novelty And Significance:** Not applicable
**Recommendation:** 1
**Confidence:** 3

**Main Review:**

In the first section, the paper motivates the moments penalization by showing that variance penalization leads to tighter confidence intervals (Maurer and Pontil, 2009). In the context of learning, the goal then becomes to find a
$\theta \in \Theta$ that minimizes the following (focusing on second moments for simplicity):
$$ E_n [\ell(X; \theta)] + \lambda V_n(\ell(X; \theta)), \qquad \qquad (1) $$
where $E_n$ and $V_n$ denote empirical mean and covariance respectively of $X \sim P_n$, the empirical distribution.
Hence the objective is to find a choice of weights $W(X;\theta)$, that may depend on $\theta$, so that the following holds:
$$ E_n[W(X;\theta) \ell(X; \theta)] =  E_n [\ell(X; \theta)] + \lambda V_n(\ell(X; \theta)) \qquad \qquad (2).$$

Theorem 1 shows that if one chooses $W(X;\theta) = 1 + \lambda (\ell(X;\theta) - E_n[\ell(X;\theta)] )$, then (2) above holds.

At this point, this result is not useful unless it is easy to minimize $E_n[W(X;\theta) \ell(X; \theta)]$:  as Theorem 1 shows equality with (1), one could have directly tried optimizing (1) above. This is where Lemma 2 comes in (preserving convexity).

However, the paper just drops the dependence on $\theta$ everywhere in the paper: If $W$ were to depend on $\theta$, which it does, then I do not think Lemma 2 is true, i.e., $E_n[W(X;\theta) \ell(X; \theta)]$ might not be convex in $\theta$. As $W$ clearly depends on $\theta$, I do not even understand what Lemma $2$ means when $W$ does not depend on $\theta$ --- convex in which parameter? I would appreciate if authors can clarify if my arguments are incorrect.

After hiding this dependence on $\theta$, the theoretical results in the paper ---Theorem 1, Lemma 2, Theorem 3 --- become straight-forward.
Even in the experiments section, as noted on Page 4, Section 2, the operations for computing weights are not part of the computation graph, and it is not clear what Algorithm 1 is minimizing.

It is possible that Algorithm 1, by detaching weights from the computation graph, is approximately optimizing the desired objective of (1), but I don't see any immediate connection at this moment. Investigating this connection might be a possible research direction for future but it is beyond the scope of the submitted paper.

Though paper claims that it builds and extends the works of Duchi and Namkoong (2019) and Li et al. (2021), this limitation of the present paper is not present there.

I would be happy to increase my scores if my understanding of the results in the paper is flawed, and authors can clarify their contributions.


## Minor Comments

1. Since the goal is to penalize by empirical variance, first section should cite the results from (Maurer and Pontil, 2009) who showed that penalizing from empirical variance also works (instead of true variance as is cited in Equation (1) from Hoeffding's inequality).
3. Proposition 7 is a basic fact in probability and certainly not attributed to Duchi and Namkoong (2019).



**Summary Of The Paper:**

The paper studies the role and implications of weighted empirical risk minimization, where one can also choose the weight of each sample instead of fixing it to be equal.
The paper then shows that specific choices of weights lead to variance penalization and higher-order moments penalization (Theorems 1 and 3). Lemma 2 and 4 study the regimes for which these chosen weights also preserve the convexity of the original loss functions.
This framework is then empirically validated through multiple experiments.

**Summary Of The Review:**

In my current understanding, the paper seems to study a simplistic setting (and hence the theoretical results of the paper are immediate) and it is not clear to me how the results/framework extends to the general setting. Hence, I do not think the paper is fit for ICLR in its current form. I will be happy to increase my score if my understanding is incorrect.

---

> ### Author Response · Authors · 2021-11-14
> **We revised the notation to remove the confusion**
>
> We thank the reviewers for the comments. We followed the reviewer’ suggestions closely and updated the paper. We hope the changes to the revised manuscript remove the confusion.
>
> We will first summarize the changes and then provide point-by-point answers to all received comments.
>
> ### Main changes
>
> - We included a new section entitled “Notations” containing the conventions and notations.
> - We updated the notations used in the theoretical results concerning the moments penalization and robustness.
> - We removed the confusing “detach from computational graph” part of the algorithm.
> - We revised Appendix A to follow the updated notation.
>
> ### Point by point answers
>
> - In response to “As W clearly depends on theta, I do not even understand what Lemma 2 means when W does not depend on theta --- convex in which parameter? I would appreciate if authors can clarify if my arguments are incorrect.”
>
>     *We hope the new section on notation along with the revised version of Lemma 2 clarifies this confusion. We define the loss function as $\ell:\mathcal{V} \times \mathcal{Y} \rightarrow [0, \infty)$ and note that this is a function that gives a penalty $\ell(v,y)$ when the model predicted the value $v$ and label $y$ was observed. As we are interested in optimizing the model output value $v$ that minimizes the penalty, we will focus our investigation on loss functions $\ell(v, y)$ that are convex in $v$ and seek to preserve the convexity when optimizing the weighted mean.*
>
>     *As we show in the proof of Lemma 2, the weight function is convex in $v$ if $\lambda$ is positive. However, the product $w(v, y)\ell(v, y)$ is not convex when $\lambda > 1/(\mathbb{E}[\ell ] - \min \ell )$ which gives us the limits of the $\lambda$ interval when the convexity is preserved.*
>
>     *To illustrate the implications of Lemma 2, we plotted $\lambda_{max}$ for the toy problem illustrated in Figure 1 inspired from Duchi and Namkoong (2019). Given $X \sim \mathcal{U}(\{-2,-1,0,1,2\})$ and $\ell(\theta; X) = |\theta - X|$ we show $\mathbb{E}[\ell(\theta, X)]$ , $\mathbb{V}[\ell(\theta, X)$, and $\lambda_{max}$. [Anonymized link to figure](https://ibb.co/mcWBp0x)*
>
>
> - In response to “the operations for computing weights are not part of the computation graph, and it is not clear what Algorithm 1 is minimizing”
>
>     *We removed the statement related to the computational graph as it is an implementation detail and will include it as a comment in the source code. Modern ML frameworks such as Tensorflow or Pytorch automate the computation of the gradient by tracking all the operations executed with the data and building a computational graph. This graph is later used during the backpropagation stage to calculate the gradients. Since weights are calculated using the values returned by the loss function, the weights will also be part of the computational graph. While this should not be an issue, we obtained mixed results with Pytorch. Probably this is due to internal implementation detail. Forcing the framework to treat the weights as constants (detaching from the computational graph) solved the issue in this situation. To avoid this confusion we removed those statements from the paper and included them in the source code as a comment.*
>
> - In response to “Since the goal is to penalize by empirical variance, first section should cite the results from (Maurer and Pontil, 2009) who showed that penalizing from empirical variance also works (instead of true variance as is cited in Equation (1) from Hoeffding's inequality). ”
>
>     *We agree with the reviewer and updated Equation (1) to reflect the result of Theorem 4 of Maurer and Pontil (2009) which uses the empirical variance instead of the theoretical one.*
>
> - In response to “Proposition 7 is a basic fact in probability and certainly not attributed to Duchi and Namkoong (2019).”
>
>     *We agreed with the reviewer and updated the corresponding statement which now reads “First, we present a proposition used in the proof of Theorem 1.”*

---

> > ### Comment · Reviewer_ndU8 · 2021-11-24
> > **Original opinion unchanged**
> >
> > I thank the authors for their thoughtful response.
> >
> > + The new notation section helps but I would still recommend using the full notation in the theorem statements.
> >
> > + Still the convexity of the loss in parameter v in Lemma 2 does not seem particularly useful at this point to me unless we can show that new loss function (including $w$) satisfies nice properties when seen as a function of $\theta$.
> >
> > + Regarding the computational graph:
> >  + I would like to clarify that the line regarding the computational graph was NOT confusing, and I would strongly suggest keeping it (or something along these lines) in the main paper as it is an important implementation detail.
> >
> >  + My point was that the Algorithm proposed in this paper is not minimizing $E[w(\theta)\ell(\theta)]$ because it is performing gradient update by only using $E[w(\theta)\nabla \ell(\theta)]$. To minimize $E[w(\theta)\ell(\theta)]$, by chain rule there would be another term as well. Thus it is not clear what loss function the Algorithm is trying to optimize.

---

> > > ### Author Response · Authors · 2021-11-25
> > > **We thank the reviewer for the feedback**
> > >
> > > We thank the reviewer for the response, unfortunately, the response came a few hours after the second stage of discussion ended and we are no longer able to update the manuscript. However, we will address all of the confusions in the camera-ready version.
> > >
> > > The reviewer is right that we are "performing gradient update by only using $\mathbb{E}[w(\theta)\nabla \ell(\theta)]$" and this is precisely our intention. The weights have the role to modify the empirical distribution with the respect to which the expectation is taken (Equation 5 in the latest version of the manuscript) similarly to the work of Duchi & Namkoong (2019).
> > >
> > > We are still puzzled by the given score and we hope the reviewer would clarify more the reasoning behind it.

---

> > > > ### Comment · Reviewer_ndU8 · 2021-11-25
> > > > **Further details**
> > > >
> > > > I thank the authors for their quick reply.
> > > >
> > > > My concern is the following: the paper motivates the choice of weights $w(X;\theta)$ so that $\mathbb{E}[w(X;\theta)\ell(X;\theta)]$ equals the standard loss + moment penalization. And thus the goal is to minimize $\mathbb{E}[w(X;\theta)\ell(X;\theta)]$ (so that we can actually minimize the regularized loss).
> > > >
> > > >  Running gradient descent on $\mathbb{E}[w(X;\theta)\ell(X;\theta)]$ would update $\theta$ by $\nabla(\mathbb{E}[w(X;\theta)\ell(X;\theta)])$, which by chain rule (and "taking" the gradient inside the expectation) should be $\mathbb{E}[w(X;\theta)\nabla\ell(X;\theta)] + \mathbb{E}[\ell(X;\theta)\nabla w(X;\theta)]$. However, the proposed algorithm is only using the first term.
> > > > Thus Algorithm 1 is not a gradient descent and it is not clear to me that the proposed algorithm is indeed minimizing $\mathbb{E}[w(X;\theta)\ell(X;\theta)]$ as claimed.

---

### Official Review · Reviewer_i96S · 2021-11-05

**Correctness:** 4
**Technical Novelty And Significance:** 2
**Empirical Novelty And Significance:** 2
**Recommendation:** 3
**Confidence:** 4

**Main Review:**

The problem studied is interesting, but the provided answer seems very limited to me, especially at the theoretical level. In particular,

**I am a bit puzzled by the negative $\lambda$**:
- it seems to me that it allows to do pretty much everything. For instance, the initial motivation is provided by eq. 1 (on another note, the link between eq. 1 and minimizing eq. 3 could be better explained, as it is not immediate to me). However, eq. 3 only makes sense if $\lambda$ is positive, but Section 3 is all about negative $\lambda$.
- the sentence "Case in which we consider them as unlearned samples and amplify their impact or as outliers and
suppress their impact on the model" is very revealing to me, as it shows you can do one thing and its opposite, but the most difficult question remains: is the data point an outlier or unlearned? How can you answer this?
- what is the motivation/intuition/incentive for minimizing eq. 4 instead of the standard sum of losses?
- it is a bit disappointing that so much effort is put in Section 2 to show convexity properties, that are totally omitted in Section 3. The weights used are actually completely different from the family studied in Section 2, breaking the coherence of the paper
- overall, my feeling is that we have a motivation for variance penalization (eq. 1), but the approach proposed is the complete opposite as a negative $\lambda$ is used. As for higher order moments, I struggle to see any similar justification...

**About practical aspects**:
- the weights depend on expectations that are unknown. Although it is specified that the results stated still hold with the empirical distribution, then one has to relate the minimized empirical quantity to the true risk, which seems nontrivial.
- as highlighted by the authors, considering higher order moments also multiply the number of hyperparameters. I wonder to what extent it is not always possible to find a combination of (positive or negative) $\lambda_i$ such that there is an improvement
- do authors have any convergence guarantee about Algorithm 1?
- about Algorithm 1: why not defining $\tilde{\ell}_i = w_i \cdot \ell_i$ and perform Gradient Descent on $\tilde{\ell}$?

**Two relevant papers/ideas**:
- "Robust multivariate mean estimation: the optimality of trimmed mean" by Lugosi & Mendelson (2021). One could imagine a weighted ERM procedure with 0/1 weights determined by the Trimmed Mean approach on the loss values. How would this approach compare to the one developed in this paper? Some thresholding ideas seem to be shared
- "Robust classification via MOM minimization" by Lecué et al. (2020). MoM-minimization can be seen a weighted ERM, with nontrivial 0/1 weights that depend on the median operator, see in particular Algorithm 1 therein. MoM-minimization has been recently shown to be robust to the corruption model presented at the beginning of Section 3 ("Generalization Bounds in the Presence of Outliers: a Median-of-Means Study" by Laforgue et al. 2021). How can it be related to the weighted approach proposed?

**Minor comments**:
- notation inconsistency: $n$ refers both to the number of samples (eq. 1, Algorithm 1, empirical expectation above Alg. 1), the highest order for the moments (Theorem 3), and the dimension of the input set (Lemma 4)
- "is bounded by the variance" --> "is bounded in terms of the variance"
- *with respect to* (2nd paragraph)
- $W$ is not defined in eq. 2
- the second sentence of Section 2 is not understandable
- $\lambda_{min}$ in Figure 1 is never defined
- I find the $W$ notation a bit confusing (in Theorem 1 for instance), as capital letters usually refer to matrices. Why not using $w = 1 + \lambda(\ell - \mathbb{E}[\ell])$, $w\ell$ (or $w \cdot \ell$) and $w(x)\ell(x)$?
- in Theorems 1/3, replace "taking the weighted mean of $\ell$ with weights $W$ is equivalent to" by "we have"
- Theorem 1 should be capitalized, i.e., not theorem 1 (Lemma 2, above Theorem 3)
- "changes with each iteration of the optimization algorithm" is not clear at all (at least at this point where Alg. 1 is not yet introduced)
- about the comments on the interval in Lemma 2, you could add that when $\mathbb{E}[\ell] = \min \ell$, you actually have $\ell$ constant, i.e., $\mathbb{V}(\ell) = 0$, and the admissible interval is $\mathbb{R}^+$, which makes sense since $\mathbb{V}(\ell) = 0$ and the "penalized" criterion is actually always equal to the original one, which is convex
- "the three central points aligned across a horizontal line" is not clear: the other cluster seems to have much more that 3 points, which ones are they? why are other points present?
- Figure 2: $\lambda_0$ is not defined, it starts at $\lambda_1$ in the given definition
- in Lemma 4, I would definitely use $w(x)\ell(x)$, to show that $W$ depends on $x$
- could be nice to highlight that Lemma 4 does not generalize Lemma 2, as you assume here $p$ to have positive values
- ~When~ clipping negative weights to 0 ~it~ prevents...
- Agorithm 1: $\mathfrak{L}$ is not defined, nor $k$ and $\gamma$
- Eq. 5 is pretty unclear: what is $k$ here? the loss is not taken w.r.t. to $i$, not $y_i$? Also there is always a $C$ satisfying the equation (the one equal to the left hand side), so what is the condition upon? Should $C$ be independent of some parameters?
- Lemma 5: Let $\ell, W$ **be**. Also $\ell$ must be positive
- Proof of Lemma 2: using $p(t)$ instead of $p(x)$ would be less confusing. Also, you provide a proof for a general $p$ while we have $p(t) = 1 + \lambda t$ at the beginning, this is also confusing. In particular the inequality $p(a) - p(b) \ge \lambda (a- b)$ holds with equality for instance. Next, we use Proposition **9**
- Proof of Lemma 4: it seems that you need $p$ to be (sub)differentiable

**Summary Of The Paper:**

This paper studies weighted Empirical Risk Minimization (ERM), where the weight at data point $(x_i, y_i)$ is a (polynomial) function of the loss value $\ell(f(x_i), y_i)$. For affine functions, the authors show that it is equivalent to perform variance penalization (Theorem 1 and Lemma 2). Polynomials with higher degrees involve higher order moments of the losses (Theorem 3 and Lemma 4). The authors also propose an iterative algorithm to perform the weighted ERM (Algorithm 1). In Section 3, a different choice of weights is proposed, that ensure the convexity of the cross-entropy criterion despite a negative $\lambda_1$, and experiments are presented.

**Summary Of The Review:**

Interesting problem but insufficient theoretical contribution. The provided insights are made confused by the fact that both positive and negative $\lambda$ are considered, with completely opposite effects. The technical derivations are basic computations.

---

> ### Author Response · Authors · 2021-11-25
> **We thank the reviewer fo the feedback**
>
> We thank the reviewers for the detailed feedback and constructive comments. We followed the reviewer’s suggestions closely and updated the paper. We hope the changes to the revised manuscript address all the issues. To access the pdfdiff between the initial submission and the most updated version, please click the “Show Revisions” link between the title and abstract (requires login) followed by the “Compare Revisions” button and select the first (imported: 22 Nov 2021) and third (modified: 05 Oct 2021) entries and click on the “View Differences” button located on the top right to see the changes highlighted.
>
> We will first summarize the changes and then provide point-by-point answers to all the comments. The main changes are as follows:
> - We reported new experimental results of penalizing higher-order moments.
> - We included a paragraph regarding the convergence of the algorithm.
> - We studied and applied the methods from the suggested papers.
> - We included a new section entitled “Notations” containing the conventions and notations.
> - We updated the notations used in the theoretical results concerning the moments penalization and robustness.
> - We incorporated all the suggestions from the Minor comments section.
>
> Point by point answers:
>
> - In response to “It seems to me that it allows to do pretty much everything. For instance, the initial motivation is provided by eq. 1 (on another note, the link between eq. 1 and minimizing eq. 3 could be better explained, as it is not immediate to me). However, eq. 3 only makes sense if $\lambda$ is positive, but Section 3 is all about negative $\lambda$.”
>
> The reviewer is right in that the framework is flexible and can be applicable to many scenarios. One interpretation of the weighted mean trick is that the loss function can be “bent” upwards or downwards to increase or decrease the impact of some samples on the model parameters. As stated in the introduction, the two papers influential for this study are Duchi & Namkoong (2019) and Li et al. (2021). The work of Duchi & Namkoong (2019) investigates the variance penalization in the context of bias-variance tradeoff for tightening the Bernstein bound (1). On the other hand, the work of Li et al. (2021) in addition to the bias-variance tradeoff also explores optimizing for robustness when $t < 0$. Our work is an extension to both as our approach allows us to independently penalize any number of higher-order central moments. We specifically chose to investigate the direct link between robustness and variance penalization (maximization since $\lambda < 0$ in this case) which we believe would be of high interest to the scientific community.
>
> - In response to “The sentence "Case in which we consider them as unlearned samples and amplify their impact or as outliers and suppress their impact on the model" is very revealing to me, as it shows you can do one thing and its opposite, but the most difficult question remains: is the data point an outlier or unlearned? How can you answer this?
>
> We do not want to put limitations on whether the data point is an outlier or unlearned. The weighted mean trick introduced in the paper provides the flexibility to treat data points as outliers or unlearned. For example, the right plot in Figure 1 shows 3 possible classification boundaries. The boundary corresponding to $\lambda > 0$ was obtained by increasing the weights of samples close to the boundary and thus it follows the orientation of the samples indicated by the square-shaped data points. On the other hand, the boundary corresponding to $\lambda < 0$ resulted from increasing the weight of samples located further away from the boundary, and thus its orientation follows that of the samples indicated by the circle-shaped data points. In Section 3, we use negative variance penalization to lower the impact of both misclassified samples and those close to the boundary on the classification boundary.
>
> - In response to “what is the motivation/intuition/incentive for minimizing eq. 4 instead of the standard sum of losses?”
>
> There are two major advantages of using the weighted mean instead of the sum of moments objective:
> It provides an algebraic interpretation, and as a result, the impact of what samples will be amplified or suppressed becomes clear as illustrated in Figure 4 from Appendix C.
> By clipping negative weights to 0, the convexity of the objective can be preserved if the polynomial function used to compute the weights is convex and non-decreasing as shown by Lemma 4. On the contrary, the convexity of the objective cannot be preserved when using the sum of moments.

---

> > ### Author Response · Authors · 2021-11-25
> > **We thank the reviewer fo the feedback**
> >
> > - In response to “it is a bit disappointing that so much effort is put in Section 2 to show convexity properties, that are totally omitted in Section 3. The weights used are actually completely different from the family studied in Section 2, breaking the coherence of the paper”
> >
> > We thank the reviewer for sharing this perspective with us. As stated before, we tried to connect the variance penalization with robustness, in particular, variance maximization since $\lambda < 0$. This breaks convexity and the results of the “Moments Penalization” section help understand that. Moreover, we developed a convex version of the moments penalization for scenarios that require convexity. Again, our goal was to directly connect variance penalization with robustness which we believe would be of high interest to the scientific community.
> >
> > - In response to “overall, my feeling is that we have a motivation for variance penalization (eq. 1), but the approach proposed is the complete opposite as a negative is used. As for higher order moments, I struggle to see any similar justification…”
> >
> > Equation 1 serves as a justification for variance penalization with which more people are familiar with. However, as stated before, we specifically chose to investigate the direct link between robustness and variance penalization (maximization since $\lambda < 0$ in this case) which we believe would be of interest to the scientific community. With respect to the higher-order moments, penalizing them gives more flexibility in shaping the loss function as illustrated in Figure 4 and Appendix C and where we included the robustness results when penalizing higher-order moments.
> >
> > - In response to “the weights depend on expectations that are unknown. Although it is specified that the results stated still hold with the empirical distribution, then one has to relate the minimized empirical quantity to the true risk, which seems nontrivial.”
> >
> > When applied in practice, the method penalizes the empirical variance and the empirical higher-order central moments. The connection between the empirical values and the true ones is an interesting research direction and will be investigated in the future.
> >
> > - In response to “as highlighted by the authors, considering higher order moments also multiply the number of hyperparameters. I wonder to what extent it is not always possible to find a combination of $\lambda_i$ (positive or negative) such that there is an improvement”
> >
> > As the reviewer correctly stated as part of “Minor comments”, when a higher-order central moment is 0, penalizing it has no effect. However, in practice, due to the limited number of samples and numerical precision, the empirical mean does not exactly match the theoretical one. Thus, even if the loss values follow a normal distribution, the computed skewness and kurtosis will not be 0 and thus these central moments are technically penalizable.
> >
> > - In response to “do authors have any convergence guarantee about Algorithm 1?”
> >
> > We included convergence details at the end of Section 3 “Moments Penalization”. The moments penalization problem falls under the class of distributional robust stochastic programs where the weights $w$ alter the empirical distribution with respect to which the expectation is taken. Informally, the algorithm will converge if changes in its parameters will cause minor changes in the distribution with respect to which the expectation is taken, and with each step, it will approach the optimum distribution. The moments penalization factors $\lambda_i$ for $i \geq 2$ determine how much the distribution changes when the loss values change. Small values of the penalization factors will keep the distribution in the neighborhood of the empirical one. The exact values depend on the empirical distribution of the data and the choice of both the model and loss function.
> >
> > - In response to “about Algorithm 1: why not defining $\tilde{\lambda}_i = w_i \cdot \ell_i$ and perform Gradient Descent on $\tilde{\lambda}_i$?
> >
> > Indeed, this is what the algorithm is optimizing and as a result, we simplified the notation. We feel the comment related to the computational graph created confusion. We removed the statement related to the computational graph as it is an implementation detail and we will include it as a comment in the source code. Modern ML frameworks such as Tensorflow or Pytorch automate the computation of the gradient by tracking all the operations executed with the data and building a computational graph. This graph is later used during the backpropagation stage to calculate the gradients. Since weights are calculated using the values returned by the loss function, the weights will also be a part of the computational graph. While this should not be an issue, we obtained mixed results with Pytorch. This is probably due to internal implementation details. Forcing the framework to treat the weights as constants (detaching from the computational graph) solved the issue in this situation.

---

> > > ### Author Response · Authors · 2021-11-25
> > > **We thank the reviewer fo the feedback**
> > >
> > > - In response to “Two relevant papers/ideas:”
> > >
> > > We thank the reviewer for all the suggested papers, we studied and cited the papers. To answer how the suggested approaches can be related to the current work, we see two possible scenarios:
> > > Apply the proposed approaches to calculate a robust mean, then use the robust mean value in the weights formula. As explained below, the suggested methods were not designed for a high level of contamination with outliers. However, even without the theoretical guarantees of the suggested methods, there might be a practical advantage to using a robust mean which will be explored in the future.
> > > Train the model by minimizing a robust mean. Similar to the previous scenario, the suggested methods were not designed for the noise levels considered in the current work. For example, when $\eta=0.2$ and assuming large $N$, the quantile $\epsilon$ of the univariate mean estimator from Lugosi & Mendelson (2021) is at least 1.6, a value outside the range $(0, 1]$. For MoM (Lecué et al. 2020 and Laforgue et al. 2021), the condition that the number of blocks $K$ is greater than $2n_o$, where $n_o$ is the number of outliers, cannot be satisfied for $\eta = 0.2$ as the resulting block size is 2.5 samples, not large enough to reliably estimate the empirical mean. However, we still tried to minimize a robust mean calculated using the univariate mean estimator from Lugosi & Mendelson (2021) where we used a weight function to reproduce the behavior of the $\phi_{\alpha, \beta}$ function. Unfortunately, the model did not converge and required a deeper investigation which we could not complete within the two weeks period.
> > > However, those methods sparked our interest and we will continue investigating them in the future.
> > >
> > > - In response to “Minor comments” section
> > >
> > > We thank the reviewer for all the suggestions and we agreed with all the proposed changes. We implemented all the suggestions from the minor comments into the paper.

---

> > > > ### Comment · Reviewer_i96S · 2021-11-27
> > > > **Response**
> > > >
> > > > I thank the authors for their extensive response. The revisions provided improve the paper in my opinion.
> > > >
> > > > Still, in absence of any theoretical hint on how to choose the $\lambda_i$ to improve ERM, I really feel that the empirical improvement is only due to the tuning (the best values found will always be better than $\lambda_i = 0$, which is the benchmark used). Therefore, this cannot be regarded as a proof of validity. This is even more true if you allow to consider both positive and negative values (since they have opposite effects, either one of them will improve upon standard ERM). I would recommend to only focus on negative weights, and to mention the variance penalization as a different problem.

---

### Decision · Program_Chairs · 2022-01-20

**Decision:**

Reject

**Comment:**

The paper investigates weighted empirical risk minimization where the weights on an example in the training set is given by a polynomial function evaluated on the loss on the given example. Authors show that the choice of the weighting function induces a data-dependent variance penalization in the training objective. Authors present an algorithm for weighted ERM and empirical results to support their claims. While the problem setting is broadly relevant and the approach the authors take in this paper is interesting, several questions remain unanswered. First, the authors argue that variance penalization helps but do not compare with other regularized ERM approaches. Second, it is not clear if the proposed algorithm is indeed gradient descent on the weighted ERM objective as pointed out by one of the reviewers. Finally, the writing can be improved with more emphasis on the novelty and significance of the contributions. I believe the initial comments from the reviewers has already helped improve the quality of the paper. I encourage the authors to further incorporate the feedback and work towards a stronger submission.